# HAM: Hierarchical Adapters Merging for Scalable Continual Learning

## Abstract

Continual Learning allows models to acquire knowledge incrementally, but is challenged by catastrophic forgetting, a phenomenon in which the learning of new tasks disrupts previously acquired knowledge. Although large pre-trained models can partially mitigate forgetting by leveraging their existing knowledge and over-parameterization, they often struggle when confronted with novel data distributions. Parameter-Efficient Fine-Tuning (PEFT) methods, such as LoRA, enable efficient adaptation to new data. However, they still face challenges in scaling to dynamic learning scenarios and long sequences of tasks, as maintaining one adapter per task introduces complexity and increases the potential for interference. In this paper, we introduce Hierarchical Adapters Merging (HAM), a novel framework that dynamically combines adapters from different tasks during training. For each experience, HAM trains a low-rank adapter along with an importance scalar, then dynamically groups tasks based on adapter similarity. Within each group, adapters are pruned, scaled and merged, facilitating transfer learning between related tasks. Extensive experiments on three vision benchmarks demonstrate that HAM surpasses state-of-the-art methods, achieving up to 4% accuracy improvement over the best baseline and nearly doubling efficiency in both training and inference, with particularly strong advantages as the number of tasks increases.

## 1 Introduction

Continual Learning (CL) aims to build models that can learn incrementally from sequences of tasks while retaining previously learned knowledge, reducing the phenomenon of *catastrophic forgetting*. The emergence of large pre-trained models has introduced new alternatives, which are nonetheless highly costly to retrain or fine-tune for each learning experience, which make the development of more efficient approaches essential for feasibility. **Parameter-Efficient Fine-Tuning** (PEFT) methods (Han et al., 2024) tackle this issue by adapting only a small subset of the weights of the model or introducing a limited number of trainable parameters, while keeping the base model frozen. Among PEFT techniques, **Low-Rank Adaptation** (LoRA) (Hu et al., 2021) has emerged as a popular choice due to its simplicity and effectiveness. However, LoRA, as the other PEFT methods, is optimized for static learning scenarios, where the objective is to achieve the highest possible performance on a single task. Conversely, in CL the focus in shifted more towards over-time learning and knowledge retetion. In this setting, the standard LoRA approach falls short.

Similar to classical CL methods, PEFT-based approaches face three key challenges: (i) preventing catastrophic forgetting of previously learned tasks, (ii) enabling knowledge transfer between related tasks and (iii) efficiently allocating parameters while maintaining scalability. Previous approaches typically address (i) and (ii) by either storing separate adapter modules for each task, requiring task identifiers at inference, or by complex parameter isolation strategies that limit knowledge transfer. More involved methods (Liang & Li, 2024; Wu et al.) try to improve the stability–plasticity trade-off by subspace reparameterization or decoupled magnitude/direction learning. Nonetheless, they still treat adapters independently throughout training and inference, preventing shared learning or adaptive reuse. Alternatively, post-hoc merging (Marczak et al., 2024; Yadav et al., 2023; Yu et al., 2024; Ilharco et al., 2023; Coleman et al., 2024) consolidates adapters after training, limiting knowledge transfer during learning. These methods also depend on manually chosen merging coefficients, which can become cumbersome and suboptimal—*especially as the number of tasks grows.*

---

*Equal contribution

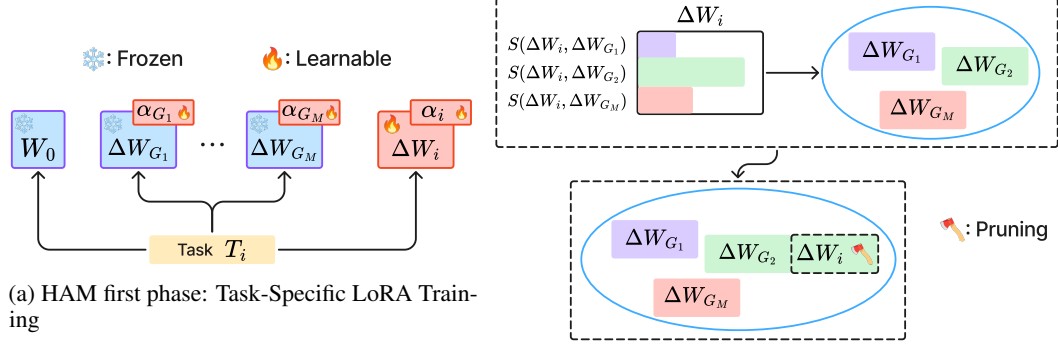

(a) HAM first phase: Task-Specific LoRA Training

(b) HAM second phase: Adapters Grouping

Figure 1: Illustration of the HAM method. Figure 1a: a new task-specific LoRA adapter $\Delta W_i$ is trained, alongside its importance factor $\alpha_i$ and $\alpha_{G_j}$, one for each of the $M$ groups, where $M \ll$ number of tasks. Figure 1b: the adapter $\Delta W_i$ is associated with the most similar group adapter. After the association, $\Delta W_i$ is pruned and concatenated with the selected group adapter.

To address these limitations, we propose *Hierarchical Adapters Merging (HAM)*, a continual LoRA-based method that dynamically groups and merges adapters as tasks unfold. For each new task $t$, HAM (i) learns a task-specific LoRA adapter with an importance weight $\alpha_t$, (ii) clusters related tasks and concatenates their adapters, and (iii) merges groups using a specialized algorithm. This hierarchical process controls the number of stored modules, retains prior-task knowledge to mitigate forgetting, and promotes positive transfer among similar tasks. Importantly, the dynamic grouping mechanism is agnostic to task ordering, making HAM more adaptive and scalable over long task sequences, outperforming single-shot and rigid merging baselines. We evaluate our method on standard CL benchmarks, putting the emphasis on longer sequences of tasks, a demanding yet realistic setting that has received limited attention in the CL community. This scenario better reflects real-world conditions, increasing the models "lifetime" and pushing the field closer to a **true lifelong learning agent**.

To summarize, in this paper we introduce **Hierarchical Adapters Merging for Low-Rank Adaptation (HAM)**, a novel approach that addresses the challenge of continually learning over long sequences of tasks through a combination of task-specific adaptations, importance-weighted pruning, and hierarchical merging. Our key contributions are as follows:

- We propose a scalable CL methodology for foundation models, leveraging PEFT techniques to ensure efficiency, and an adaptation mechanism that assigns learnable importance parameters to task-specific LoRA modules;
- We design a hierarchical group-based merging strategy that promotes knowledge transfer across tasks, mitigates catastrophic forgetting, and guarantees high performance over long task sequences;
- We conduct extensive experiments on diverse benchmarks, showing state-of-the-art performance in dynamic CL scenarios with long task sequences.

Through a comprehensive experimental evaluation (Sec. 4.2), we demonstrate that our method outperforms previous PEFT-based approaches, achieving a 4% increase in accuracy and 9% decrease in forgetting comparing to the strongest baseline, while being nearly twice as fast in both training and inference. To gain deeper insights into HAM's behavior, we conduct extensive ablation studies (Sec. 4.3), which highlight the effectiveness of our hierarchical, two-phase merging strategy. These analyses confirm that our approach substantially mitigates task interference, enhances knowledge retention, and preserves scalability and efficiency even over longer task sequences.

## 2 RELATED WORK

**Continual Learning** CL approaches are typically grouped into three categories: (i) regularization-based methods (Kirkpatrick et al., 2017; Zenke et al., 2017), (ii) replay-based methods (Rebuffi et al., 2017; Chaudhry et al., 2019) and (iii) parameter isolation methods (Mallya & Lazebnik, 2018; Rusu et al., 2016). The rise of large pretrained models has driven the adaptation of CL approaches to Transformer-based architectures (Vaswani et al., 2017). Recent work has increasingly explored how

to leverage these models for CL, highlighting both the challenges and opportunities. On one hand, the rich representations and features extraction abilities captured during pre-training can promote positive forward transfer during sequential learning. Conversely, the sheer number of parameters complicates the fine-tuning of the model, making it prohibitively expensive and often impractical.

**Parameter Efficient Fine-Tuning** These techniques allow large pre-trained models to be adapted to downstream tasks by only modifying a small set of parameters (Houlsby et al., 2019), enabling the learning of new tasks with a much higher efficiency than full fine-tuning. Low-Rank Adaptation (LoRA) (Hu et al., 2021) freezes the pre-trained model weights and injects trainable low-rank decomposition matrices into each layer. For a pre-trained weight matrix $W_0 \in \mathbb{R}^{d \times k}$, LoRA parameterizes the update $\Delta W$ as the product of two low-rank matrices:

$$W = W_0 + \Delta W = W_0 + BA \tag{1}$$

where $B \in \mathbb{R}^{d \times r}$, $A \in \mathbb{R}^{r \times k}$, and the rank $r \ll \min(d, k)$. This significantly reduces the number of trainable parameters from $d \times k$ to $r \times (d + k)$. Different LoRA-based techniques have been adapted for CL to enable efficient adaptation over time. *InfLoRA* (Liang & Li, 2024) proposes an interference-free low-rank adaptation method that reparameterizes pre-trained weights within a subspace designed to minimize interference between tasks. *SD-LoRA* Wu et al. focuses on dynamic adaptation which decouples magnitude and direction of LoRA updates for scalable, rehearsal-free class-incremental learning. *SEMA* Wang et al. (2025) introduces a mixture-of-adapters framework, dynamically expanding the adapters set for new tasks while mitigating forgetting through task-specific routing. Nonetheless, such methods fall short when dealing with long task sequences.

An alternative is represented by *prompt-based* methods, which enable continual efficient learning of new tasks though small learnable parameters. *Learning to Prompt* (L2P) (Wang et al., 2022b) introduces a framework where a pre-trained model is guided by a set of learnable prompts stored in a memory bank. These prompts are dynamically selected based on input queries, allowing the model to adapt to new tasks without modifying the core parameters. *DualPrompt* (Wang et al., 2022a) builds upon L2P by incorporating both task-invariant and task-specific prompts. This duality enables the model to capture shared knowledge across tasks while retaining task-specific nuances. *CODA-Prompt* (Smith et al., 2023) improves prompt-based methods by using an attention-driven key-query mechanism to construct input-conditioned prompts, enhancing adaptability without sacrificing past performance. However, a key drawback of prompt-based CL methods is their limited plasticity, which restricts their ability to adapt effectively to novel tasks beyond their pre-training distribution.

**Model Merging** This offers a promising approach to CL by combining expert models to mitigate catastrophic forgetting while leveraging their diverse strengths and capabilities. In dynamic and ever-evolving domains, this enables models to expand their knowledge while minimizing the loss of prior information. Different algorithms were defined to perform such combination of multiple models. Linear merging computes a parameter-wise weighted average, without any particular technique to address interference. *TIES* (Yadav et al., 2023) removes redundant parameters and solves sign conflicts before merging the model's weights. *DARE* (Yu et al., 2024) randomly drops a portion of the parameters and rescales the remaining ones to reduce redundancy and minimize interference.

Several works leverage model merging for CL. *MagMax* (Marczak et al., 2024) fine-tunes tasks sequentially, then consolidates weights via maximum-magnitude selection, requiring no retraining but applying merging only once at the end. *Orthogonal Projection-Based Continual Merging* (Tang et al., 2025) allows sequential integration of new models, using orthogonal projection to reduce interference and a scaling factor to balance contributions. *Adaptive LoRA Merging* (Coleman et al., 2025) replaces fixed-weights combinations with dynamically computed merging coefficients, enhancing task integration. *MELoRA* (Ren et al., 2024) trains smaller LoRAs in parallel, concatenating them diagonally into a single adapter, increasing representation capacity while maintaining computational efficiency over standard LoRA.

## 3 HAM: HIERARCHICAL ADAPTERS MERGING

In this section, we introduce **HAM**. We first formalize the problem setting and the objectives of our method (Sec. 3.1). We then describe the training phase of the HAM approach (Sec. 3.2), which consists of two main stages: (i) **task-specific training** of each LoRA adapter, highlighting the important role of the $\alpha$ values; and (ii) **adapter grouping**, where the new LoRA is clustered, pruned and combined into the most similar adapter group. At inference time, HAM produces a unified model

through a final adapter merging step (Sec. 3.3). Importantly, this merging step can be executed at any point during training, without requiring the entire task sequence to be observed beforehand. An overview of the method is presented in Figure 1. Additionally, we provide details about HAM's implementation in Appendix A.2.

## 3.1 PROBLEM FORMULATION

We consider the class-incremental continual learning setting, where a model encounters a sequence of tasks $\mathcal{T} = \{T_1, T_2, ..., T_N\}$ overtime, with non-overlapping class sets. For each task $T_i$, the model receives a dataset $\mathcal{D}_i = \{(x_j, y_j)\}_{j=1}^{n_i}$, where $x_j$ represents an input sample and $y_j$ its corresponding label from the class set specific to task $T_i$. The model must learn to perform well on all encountered tasks after training sequentially on each task, without task identifiers at inference time. We employ a pre-trained model as our base model, *e.g.* a Vision Transformer, and denote its parameters as $W_0$. The objective is to adapt it to each task in the sequence using low-rank modules, while preventing catastrophic forgetting. Our goal is to design a strategy that enables effective knowledge transfer between tasks, ultimately making the method particularly well-suited for longer task sequences.

**Notation** For clarity, we introduce the notation used throughout the method description. A LoRA adapter trained on a specific task $T_i$ is denoted as $\Delta W_i = B_i A_i$, where $B_i \in \mathbb{R}^{d \times r}$ is the down-projection matrix, and $A_i \in \mathbb{R}^{r \times k}$ is the up-projection matrix. A group of LoRA adapters is represented as $G_i$, associated with a group adapter $\Delta W_{G_i} = B_{G_i} A_{G_i}$. The set of groups is written as $\mathcal{G} = \{G_1, \ldots, G_M\}$, where $M$ denotes the current number of groups. The maximum number of groups is denoted by $G_{\max}$.

## 3.2 HAM TRAINING PROCESS

### 3.2.1 TASK-SPECIFIC LORA TRAINING

For each task $T_i$, we introduce a LoRA adapter $\Delta W_i$. Alongside with the adapter, we train an $\alpha_i$ value that conveys the importance the LoRA module itself. This scalar value then serves as a scaling factor, needed to efficiently combine multiple adapters with limited interference. During the training phase for task $T_i$, the adapter $\Delta W_i$ and its corresponding scaling factor $\alpha_i$ are optimized using the dataset $\mathcal{D}_i$. In this phase, the pre-trained model weights $W_0$ and the previously learned group adapters $\{\Delta W_{G_j}\}_{j=1}^M$ (see Sec. 3.2.2), are kept frozen. Instead, while training $\Delta W_i$, the scaling factors of the group adapters $\{\alpha_{G_j}\}_{j=1}^M$ are also updated. This ensures that the relative importance of all adapters remains balanced when a new task is introduced. Therefore, the output of the model for an input $x$ is computed as:

$$h = \underbrace{W_0 x}_{\text{pre-trained model}} + \underbrace{\sum_{j=1}^M \alpha_{G_j} \Delta W_{G_j} x}_{\text{previous group adapters}} + \underbrace{\alpha_i \Delta W_i x}_{\text{current adapter}} \tag{2}$$

where we underline in red the components which are updated during the task-specific training phase. This approach guarantees that each task-specific adapter is optimized while accounting also for the behaviors and knowledge of earlier grouped LoRAs. By minimizing the amount of redundant information across adapters, it encourages each one to focus solely on task-specific features, also facilitating effective knowledge transfer between them.

As mentioned before, HAM adopts a dynamic grouping strategy to efficiently manage the increasing number of adapters. This contrast with approaches such as Wu et al., where both the number of adapters and their corresponding $\alpha$ values grows linearly with the number of tasks, leading to high computational costs. Instead, rather than retaining a separate LoRA module for each training experience, our strategy progressively clusters similar adapters, performing an initial combination phase within our hierarchical framework. Under this approach, each group $G_i$ holds a single adapter $\Delta W_{G_i}$ and a unique importance factor $\alpha_{G_i}$. By consolidating adapters in this fashion, the total number of modules to be merged at inference time is substantially reduced, minimizing task interference while preserving the expressive capacity of grouped adapters. Consequently, HAM excels in learning across extended task sequences, mitigating forgetting while maintaining high efficiency.

### 3.2.2 ADAPTERS GROUPING

**Group Association**    After training on task $T_i$ and obtaining the adapter $\Delta W_i = B_i A_i$, we compute the cosine similarity between the current adapter and previously learned group adapters, using the last LoRA layer of each adapter:

$$S(\Delta W_i, \Delta W_{G_j}) = \frac{|\langle \text{vec}(B_i A_i), \text{vec}(B_{G_j} A_{G_j}) \rangle|}{\|\text{vec}(B_i A_i)\| \cdot \|\text{vec}(B_{G_j} A_{G_j})\|}, \quad \text{where } j \in \{1, \dots, M\} \tag{3}$$

where $\text{vec}(\cdot)$ denotes vectorization. Given a similarity threshold $\tau_{\text{sim}}$, the adapter $\Delta W_i$ joins the group adapter $\Delta W_{G_j}$ if $S(\Delta W_i, \Delta W_{G_j}) \geq \tau_{\text{sim}}$. If all similarity scores fall below the threshold and the number of groups has not yet reached the limit, *i.e.* $M < G_{\text{max}}$, a new group is created for the adapter. Otherwise, when the maximum number of groups is reached ($M = G_{\text{max}}$), the adapter is assigned to the most similar group, regardless of the threshold.

Each group $G_i$ carries a single group-level scaling factor $\alpha_{G_i}$, given by the average of the individual $\alpha$ values associated to the adapters in the group. When a new adapter $\Delta W_j$ is added to a group $G_i$, the importance factor is updated as follows:

$$\begin{cases} \alpha_{G_i} = \alpha_j & \text{if } M = 0 \\ \alpha_{G_i} = \alpha_{G_i} + \dfrac{\alpha_j - \alpha_{G_i}}{|G_i| + 1} & \text{otherwise} \end{cases} \tag{4}$$

**Selective Pruning**    After selecting the most similar group, we perform selective pruning to retain only the most significant weights of the current adapter $\Delta W_i$. This pruning step is essential for maintaining high performance while reducing parameter overhead during merging. We calculate the importance of individual weights based on their magnitude. Specifically, for matrices $B_i$ and $A_i$, we retain only the top-$k\%$ weights with the highest absolute values, where $k$ is a hyperparameter. Hence, the resulting matrices $\hat{B}_i$ and $\hat{A}_i$ are defined as:

$$\begin{aligned} \hat{B}_i &= B_i \odot \mathbb{I}(|B_i| \geq \tau_B) \\ \hat{A}_i &= A_i \odot \mathbb{I}(|A_i| \geq \tau_A) \end{aligned} \tag{5}$$

where $\odot$ represents element-wise multiplication, $\mathbb{I}(\cdot)$ is the indicator function, and $\tau_B$ and $\tau_A$ are thresholds chosen such that only the top-$k\%$ elements are retained.

**Intra-Group Concatenation**    To obtain a single adapter per group, we concatenate all the LoRA modules within that group. Specifically, for the pruned adapter $\Delta \hat{W}_i$ associated with the group $G_j$, the group adapter $\Delta W_{G_j}$ is updated as follows:

$$\begin{aligned} B_{G_j} &= [B_{G_j}, \hat{B}_i] \\ A_{G_j} &= [A_{G_j}; \hat{A}_i] \end{aligned} \tag{6}$$

where $[x, y]$ denotes horizontal concatenation and $[x; y]$ denotes vertical concatenation.

We found this intra-group combination phase to be critical for the overall performance of the final algorithm. Specifically, this step is primarily necessary to reduce the total number of adapters, which has a substantial impact on the effectiveness of the subsequent merging phase. In fact, as the number of modules increases, the merging procedures face greater difficulty in computing an optimal combination of all adapters, leading to an increased interference among them and lower performance. We tested different combination techniques in this phase, like TIES (Yadav et al., 2023). However, we observed that the best results were achieved using a simple matrix concatenation to obtain the final group adapters. We attribute this to the fact that, while merging techniques such as TIES produce an output adapter with the same rank $r$ as the input LoRA modules, concatenation instead yields each $\Delta W_{G_i}$ with an expanded rank $r_{G_i} = m \cdot r$. Ultimately, this helps better preserve the features learned by LoRAs during their individual training while increasing the representation capabilities of the group adapter.

### 3.3 MODEL INFERENCE

The final stage of our method involves a global merging step, integrating all group adapters into a single module. This is necessary in class-incremental learning, where task identifiers are unavailable at inference time, making routing-based approaches unfeasible. Likewise, mixture-of-experts strategies would struggle to scale with long task sequences and introduce additional complexity for expert selection. By merging all adapters after training, HAM produces a unified model that supports inference across all tasks without requiring extra heuristics or task-specific mechanisms.

After the adapters grouping phase, each group adapter is adjusted according to its importance factor $\alpha$, which serves to mitigate potential interference arising from their combination. Consequently, equal weights are assigned to the merging algorithm, under the assumption that the $\alpha$ values provide sufficient scaling for effective integration. Therefore, the final merged adapter is computed as:

$$\Delta W_{\text{merged}} = \frac{1}{M} \sum_{i=1}^{M} \alpha_{G_i} B_{G_i} A_{G_i} \tag{7}$$

This merged adapter represents the final outcome of HAM's training phase. It encapsulates the accumulated knowledge coming from the entire training experience over time, while ensuring minimal interference across tasks. Such adapter $\Delta W_{\text{merged}}$ has a rank $r_{\text{merged}} = m \cdot r$, where $m$ is the number of adapters per group and $r$ is the rank of a single LoRA module. Therefore, our architecture expands the adapters representation space, enabling better handling of extended task sequences, while maintaining low complexity and a limited number of parameters, thanks to the considerable pruning phase performed prior to merging. The merged adapter is utilized to define the final model, starting from the pre-trained weights $W_0$. The model's updated weights are expressed as:

$$W_{\text{final}} = W_0 + \Delta W_{\text{merged}} \tag{8}$$

Essentially, HAM produces a single model that can be used seamlessly for inference across all tasks encountered during training.

In the following section, supported by a comprehensive experimental setup, we illustrate the critical role of HAM's training procedure in significantly enhancing the model's capacity to retain knowledge across a substantially larger set of tasks. Its effectiveness is demonstrated through numerous comparisons with both traditional CL baselines and more recent PEFT-based methods.

## 4 EXPERIMENTS

### 4.1 EXPERIMENTAL SETUP

We evaluate HAM on two standard CL benchmarks, namely CIFAR-100 and Tiny-ImageNet, and on the more fine-grained CUB-200 dataset, where pre-trained models usually show poor performance (Radford et al., 2021). CIFAR-100 (Krizhevsky et al., 2009) includes 60,000 images over 100 classes. CUB-200 (Wah et al., 2011) consists of 11,788 bird images across 200 categories. Tiny-ImageNet (Le & Yang, 2015) is a downsized version of ImageNet (Deng et al., 2009), with 100,000 images over 200 classes. We report two standard CL metrics: (i) **Average Accuracy (AA)**: mean accuracy over all tasks at the end of training; (ii) **Forgetting Measure (FM)**: average drop in accuracy from a task's peak to its final accuracy. Additionally, we run an efficiency analysis, showing both the training and inference times required by HAM and its competing baselines.

As for the base model, we employed a ViT-B/16 backbone pretrained on ImageNet. For HAM, we set the LoRA rank to $r = 16$, apply pruning to retain the top 60% of weights per adapter, and use $G_{\max} = 2$ task groups. Training is done using the AdamW optimizer (Loshchilov & Hutter, 2017), with a learning rate of $10^{-3}$ and batch size of 64. We compare against both standard and parameter-efficient continual learning methods. Standard baselines include fine-tuning (FT), Elastic Weight Consolidation (EWC) (Kirkpatrick et al., 2017) and Learning without Forgetting (LwF) (Li & Hoiem, 2017), implemented via Avalanche's `Naive`, `EWC` and `LwF` strategies (Carta et al., 2023). Since our method involves a merging phases, we also benchmark it against widely used merging techniques, namely Linear Merging, TIES and DARE-TIES. In this cases, we train an adapter for each task before merging the entire set using the selected algorithm, leveraging the implementation from HuggingFace `peft` library (Mangrulkar et al., 2022). Additionally, we incorporate the most recent and best-performing prompt-based and LoRA-based methods as PEFT baselines, using the

| Method | CIFAR-100 | CUB-200 | Tiny-ImageNet | Average |
|--------|-----------|---------|---------------|---------|
| Fine-Tuning | 1.90 $\pm$ 0.1 | 0.52 $\pm$ 0.1 | 2.52 $\pm$ 0.45 | 3.26 $\pm$ 0.16 |
| EWC | 1.96 $\pm$ 0.1 | 0.86 $\pm$ 0.36 | 3.12 $\pm$ 0.89 | 1.98 $\pm$ 0.32 |
| LWF | 2.35 $\pm$ 0.35 | 9.97 $\pm$ 1.3 | 4.1 $\pm$ 0.2 | 5.47 $\pm$ 0.45 |
| Linear Merging | 64.29 $\pm$ 2.98 | 45.04 $\pm$ 1.43 | 80.13 $\pm$ 0.694 | 63.15 $\pm$ 1.13 |
| TIES | 62.53 $\pm$ 0.78 | 42.31 $\pm$ 1.04 | 70.19 $\pm$ 0.12 | 58.34 $\pm$ 0.44 |
| DARE_TIES | 62.70 $\pm$ 2.04 | 42.30 $\pm$ 1.21 | 69.65 $\pm$ 0.233 | 58.22 $\pm$ 0.79 |
| L2P | 62.55 $\pm$ 2.04 | 43.25 $\pm$ 2.01 | 58.16 $\pm$ 0.69 | 54.65 $\pm$ 0.98 |
| Dual-Prompt | 61.25 $\pm$ 0.44 | 49.44 $\pm$ 1.81 | 74.83 $\pm$ 0.26 | 61.84 $\pm$ 0.63 |
| CODA-Prompt | 32.45 $\pm$ 0.47 | 41.18 $\pm$ 0.75 | 52.65 $\pm$ 0.34 | 40.1 $\pm$ 0.32 |
| SEMA | 65.86 $\pm$ 0.34 | 32.02 $\pm$ 3.16 | **83.56** $\pm$ **0.57** | 60.48 $\pm$ 1.08 |
| InfLoRA | 49.1 $\pm$ 4.4 | 36.02 $\pm$ 3.26 | 65.7 $\pm$ 0.66 | 50.27 $\pm$ 1.84 |
| SD-LoRA | 71.63 $\pm$ 0.67 | 47.56 $\pm$ 3.77 | 79.48 $\pm$ 2.09 | 66.22 $\pm$ 1.45 |
| HAM | **71.78** $\pm$ **2.24** | **55.17** $\pm$ **1.04** | 83.29 $\pm$ 0.31 | **70.08** $\pm$ **0.83** |

Table 1: Average Accuracy (%, ↑) after training on 50 tasks across three benchmarks. We also report the average forgetting across datasets for an overall comparison. All methods were trained under identical key hyperparameters (*e.g.* LoRA rank and alpha, number of epochs). Best results are shown in **bold**, and second-best results are underlined.

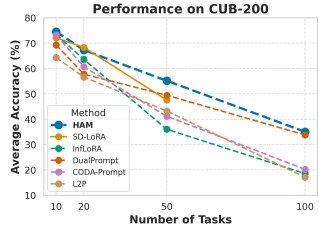

(a) Accuracy on CUB-200 with varying task counts.

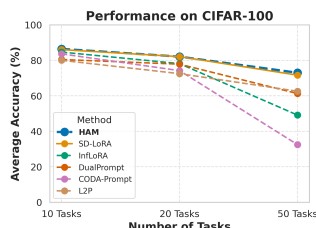

(b) Accuracy on CIFAR-100 with varying task counts.

Figure 2: Comparison of methods on CUB-200 and CIFAR-100 under different task splits.

same hyperparameters we employ for HAM. For L2P, DualPrompt and CODA-Prompt we used the implementation available in the `mammoth` library (Boschini et al., 2022). For SEMA, InfLoRA and SD-LoRA we used the official implementation from the authors.

## 4.2 MAIN RESULTS

**Accuracy** In Table 1, we report accuracy results on three datasets over a training sequence of 50 tasks. As shown in the last column, HAM significantly outperforms state-of-the-art CL methods in this challenging setting, where task sequences are longer than standard scenarios. On the simpler datasets, *i.e.* CIFAR-100 and Tiny-ImageNet, HAM achieves accuracy comparable to competing methods, nonetheless demonstrating greater consistency across benchmarks. More notably, on the fine-graned and challenging CUB-200 dataset, HAM delivers substantial accuracy improvements, where other methods struggle.

**Performance Across Different Sequence Lengths** To evaluate the effect of task sequence length, we conducted experiments with progressively longer sequences. As shown in Figure 2, HAM consistently achieves state-of-the-art performance across all sequence lengths, remaining the top-performing method in every case. Moreover, it exhibits the smallest accuracy degradation as tasks accumulate, whereas competing methods suffer sever drops. Most strikingly, Figure 2a demonstrates that HAM maintains strong performance even in the extreme case of 100 tasks, whereas one of the best baselines, SD-LoRA, fails after 69 experiences due to GPU memory limitations. These findings validate HAM's ability to achieve top-tier performance, especially when dealing with long task sequences. Thanks to its superior efficiency over the strongest competing baselines, HAM represents a compelling and scalable solution for challenging and realistic CL scenarios.

**Forgetting** Another key metric we evaluate is forgetting, which measures the ability to retain knowledge in CL scenarios. The results, reported in Table 2, are obtained from the same setting as the one presented in Table 1. We emphasize that correctly interpreting forgetting results in

| Method | CIFAR-100 | CUB-200 | Tiny-ImageNet | Average |
|---|---|---|---|---|
| Fine-Tuning | 97.63 ± 0.67 | 24.4 ± 2.86 | 97.95 ± 0.55 | 73.33 ± 1.00 |
| EWC | 97.63 ± 0.5 | 26.26 ± 4.31 | 96.62 ± 0.99 | 73.50 ± 1.48 |
| LWF | 98.72 ± 0.33 | 14.29 ± 2.46 | 95.22 ± 0.5 | 69.41 ± 0.84 |
| Linear Merging | 31.79 ± 2.11 | 29.17 ± 1.40 | 16.50 ± 1.11 | 25.82 ± 0.92 |
| TIES | 30 ± 2.04 | 29.18 ± 1.39 | 17.94 ± 20.2 | 25.71 ± 6.78 |
| DARE_TIES | 31.04 ± 2.14 | 29.10 ± 2.17 | 17.53 ± 0.34 | 25.89 ± 1.02 |
| L2P | 18.44 ± 0.64 | **5.53 ± 1.19** | 6.83 ± 0.90 | 10.27 ± 0.54 |
| Dual-Prompt | **10.58 ± 1.66** | 8.53 ± 2.05 | **5.85 ± 0.52** | **8.32 ± 0.90** |
| CODA-Prompt | 13.30 ± 0.64 | 11.78 ± 1.18 | 12.29 ± 1.98 | 12.46 ± 0.80 |
| SEMA | 14.67 ± 0.67 | 13.26 ± 1.32 | 10.86 ± 0.77 | 12.93 ± 0.56 |
| InfLoRA | 23.41 ± 3.84 | 46.26 ± 3.68 | 19.32 ± 1.82 | 29.66 ± 1.87 |
| SD-LoRA | 12.24 ± 1.45 | 33.15 ± 1.45 | 10.63 ± 4.0 | 18.67 ± 1.50 |
| HAM | 10.98 ± 0.77 | 12.94 ± 1.65 | 5.32 ± 0.45 | 9.75 ± 0.63 |

Table 2: Forgetting Measure (%, ↓) after training on 50 tasks across three benchmarks. We also report the average forgetting across datasets for an overall comparison. All methods were trained under identical key hyperparameters (*e.g.* LoRA rank and alpha, number of epochs). Best results are shown in **bold**, and second-best results are underlined.

isolation is challenging; a comprehensive understanding requires analyzing them in conjunction with overall accuracy. In fact, while DualPrompt achieves the lowest forgetting across all experiments, its accuracy performance is substantially weaker, falling about 9% behind HAM on average. In contrast, when comparing with the strongest baselines in terms of accuracy, namely SEMA, InfLoRA and SD-LoRA, HAM not only achieves higher accuracy but also exhibits consistently lower forgetting. This highlights the effectiveness of our hierarchical approach in mitigating interference during the merging phase, ultimately improving both knowledge retention and predictive performance.

**Computational Efficiency** To assess computational efficiency, we compare HAM's training and inference times with the best performing PEFT-based CL baselines. Figure 3 reports the average per-task times on CIFAR-100 with 10 tasks, with inference times measured on the final model after training. HAM is substantially faster than the competing methods, both during training and inference. Notably, HAM is extremely more efficient than SEMA and roughly twice as fast as SD-LoRA in both stages, while also delivering superior performance in terms of accuracy and reduced forgetting. This efficiency gains is largely due to HAM's hierarchical merging strategy and pruning approach, which reduce the number of active parameters and lower computational overhead throughout training and inference.

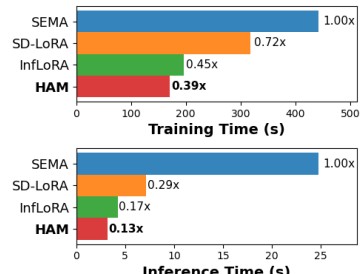

Figure 3: Training and inference times comparison. HAM is significantly faster than the strongest competing approaches.

### 4.3 ABLATION STUDIES

**Similarity-based *vs* Orthogonality Grouping** A fundamental design choice in HAM is grouping tasks by *similarity* rather than *orthogonality*. While orthogonal grouping might intuitively seem beneficial for maximizing diversity within groups, our experiments (Table 3) demonstrate that similarity-based grouping consistently yields superior performance, with the advantage becoming more pronounced as the number of tasks increases.

| Dataset | # Tasks | Similarity (AA) | Orthogonal (AA) | Advantage |
|---|---|---|---|---|
| CIFAR-100 | 20 | 82.45 % | 81.47 % | +0.98 % |
| CIFAR-100 | 50 | **68.50 %** | 64.86 % | **+3.64 %** |
| CUB-200 | 20 | 66.19 % | 64.49 % | +1.69 % |
| CUB-200 | 50 | **55.25 %** | 50.16 % | **+5.08 %** |

Table 3: Performance comparison of similarity-based vs. orthogonality-based grouping across datasets and task counts. Similarity-based grouping consistently outperforms orthogonal grouping, with larger gains as the number of tasks increases.

**Group Adapters as Effective Intermediate Models** A surprising finding in our experiments is that the intermediate group adapters, created through concatenation and pruning, consistently outperform

| Group | Task ID | Individual (%) | Group (%) | Improvement (%) |
|---|---|---|---|---|
| | Task 3 | 84.40 | 87.80 | **+3.40** |
| | Task 6 | 86.70 | 89.60 | +2.90 |
| Group 0 | Task 8 | 81.50 | 85.20 | **+3.70** |
| | Task 1 | 84.30 | 85.20 | +0.90 |
| | Task 7 | 83.60 | 86.30 | +2.70 |
| | Task 2 | 80.90 | 83.60 | +2.70 |
| | Task 4 | 78.70 | 83.50 | **+4.80** |
| Group 1 | Task 5 | 87.00 | 90.60 | +3.60 |
| | Task 9 | 90.00 | 92.60 | +2.60 |
| | Task 0 | 84.20 | 84.30 | +0.10 |
| **Average** | | **84.13** | **86.87** | **+2.74** |

Table 4: Per-task accuracy improvement with group adapters at $k = 0.6$ (40% sparsity).

their constituent individual adapters. Table 4 details the per-task gains at our selected $k = 0.6$, showing that initially weaker tasks benefit most: Task 4 improves by +4.80% and Task 8 by +3.70%. With $k = 0.6$, we reduce parameters from 5.90M to 3.54M ($1.67\times$ reduction) while achieving +2.74% accuracy improvement. This demonstrates that concatenation with moderate pruning is not merely a parameter reduction technique but an effective *knowledge consolidation mechanism*.

**Impact of the Number of Groups** Table 5 reveals that using fewer groups generally improves performance, while a similar reduction with 20 tasks yields a 0.96% improvement. This suggests that larger groups may facilitate better knowledge sharing across related tasks, though HAM remains robust across different grouping configurations with relatively modest performance variations.

**Impact of the Merging Algorithm** A crucial step in HAM is the final merging stage, where group adapters are consolidated into a single one. To evaluate its impact, we compared several merging algorithms on a 10-task scenario with rank 16 LoRA adapters (Table 6). Linear merging achieved the best accuracy, slightly outperforming TIES and clearly surpassing DARE. Notably, the simpler linear strategy outperformed more sophisticated methods, likely due to better preservation of task-specific knowledge within HAM's hierarchical structure.

**Impact of Pruning** Table 7 shows HAM's performance when varying the percentage of retained weights ($k$). AA improves as $k$ increases from 10% to 60%, after which gains plateau and performance slightly declines at $k = 80\%$. The most substantial improvements occur between 10% and 40%, suggesting that strong pruning effectively balances information retention and noise reduction. Indeed, extremely low retention ($k = 10\text{-}20\%$) causes information loss, while higher levels ($k > 60\%$) reintroduce noise. Overall, these results confirm HAM's design choice of employing strong pruning to optimize both performance and parameter efficiency.

| # Groups | # Tasks | AA (%) |
|---|---|---|
| 2 | 10 | **86.44** $\pm$ **0.30** |
| 5 | 10 | 83.25 $\pm$ 0.21 |
| 2 | 20 | **82.45** $\pm$ **0.45** |
| 4 | 20 | 78.81 $\pm$ 0.01 |

Table 5: Impact of the number of groups on CIFAR-100.

| Merging | AA (%) |
|---|---|
| Linear | **86.44** $\pm$ **0.30** |
| TIES | 85.66 $\pm$ 0.13 |
| DARE | 83.43 $\pm$ 0.07 |

Table 6: Impact of different algorithms in HAM final merging.

| $k$ | Params (Pruned) | Reduction Factor | AA (%) |
|---|---|---|---|
| 0.9 | 5.31M | $1.11\times$ | 83.12 |
| 0.8 | 4.72M | $1.25\times$ | 85.82 |
| **0.6** | **3.54M** | **1.67$\times$** | **85.83** |
| 0.4 | 2.36M | $2.50\times$ | 85.64 |
| 0.2 | 1.18M | $5.00\times$ | 84.41 |
| 0.1 | 0.59M | $10.00\times$ | 83.12 |

Table 7: Impact of pruning ratio on CIFAR-100.

## 5 CONCLUSION

This study introduces Hierarchical Adapters Merging (HAM), a novel approach to LoRA merging designed for Continual Learning. HAM follows a two-step merging procedure: first, it clusters and concatenates similar adapters; then, it scales them using group-specific importance factors before performing group-wise merging. Extensive experiments validate its effectiveness, showing superior performance compared to state-of-the-art techniques across long task sequences, reducing interference among adapters and enhancing transfer learning between tasks. In this work, we used LoRA because of its balance between learning capabilities and computational complexity. However, it is theoretically possible to use other PEFT methods, *e.g.* prompts, and it would be interesting to assess the differences in performance when changing PEFT technique. Additionally, possible future works can be directed towards a variation of such method that enables an online adapters merging.

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

# A    Appendix

## A.1    Computational Resources

All experiments were conducted on the Leonardo supercomputer at CINECA. Table 8 provides detailed specifications of the computational environment used for reproducibility.

| System | Details |
|---|---|
| **Cluster** | Leonardo @ CINECA |
| **OS** | Red Hat Enterprise Linux 8.7 (Ootpa) |
| **Booster Module (GPU)** | Atos BullSequana X2135 "Da Vinci" blades |
| **Booster Nodes** | 3456 compute nodes |
| **CPU** | 32 × Intel Ice Lake @ 2.60 GHz |
| **GPU** | 4 × NVIDIA A100 (Ampere), 64 GB |
| **RAM** | 512 GB per node |
| **DCGP Module (CPU)** | Atos BullSequana X2140 blades |
| **DCGP Nodes** | 1536 compute nodes |
| **CPU** | 2 × 56-core Intel Sapphire Rapids @ 2.00 GHz |
| **RAM** | 512 GB per node |
| **Network** | 200G HDR Infiniband Dragonfly+ |
| **Software Environment** | Python 3.11, PyTorch 2.0.1, CUDA 11.8, HuggingFace PEFT 0.15.2 |

Table 8: System configuration and environment used for all experiments.

## A.2    HAM Algorithm

In this section, we detail the HAM algorithm through pseudo-code presented in Algorithm 1, offering a clear and reproducible outline of its computational steps.

## A.3    Additional Baselines

Table 9 reports additional baseline comparisons, including joint training approaches where the model learns all tasks simultaneously. Joint training serves as an upper bound for performance, as it has access to all task data concurrently. In particular, *joint* refers to the full fine-tuning of the model on all data, while *joint LoRA* indicates the performance when training a single LoRA adapter on all tasks.

| Method | CIFAR-100 (AA) | CUB-200 (AA) |
|---|---|---|
| Joint | 83.11 ± 0.50 | 78.56 ± 2.17 |
| Joint LoRA | 86.58 ± 1.27 | 37.53 ± 1.65 |
| SEMA | 65.86 ± 0.34 | 32.02 ± 3.16 |
| InfLoRA | 49.10 ± 4.40 | 36.02 ± 3.26 |
| SD-LoRA | 71.63 ± 0.67 | 47.56 ± 3.77 |
| **HAM** | **71.78 ± 2.24** | **55.17 ± 1.04** |

Table 9: Average Accuracy (%, ↑) on CIFAR-100 and CUB-200 across 50 tasks. Joint performance represents an upper bound.

## A.4    Comparison of Merging Strategies

Table 10 compares HAM against standard model merging baselines: TIES, LINEAR, and DARE TIES on CIFAR-100 across varying task lengths. HAM consistently outperforms other methods, with

---

**Algorithm 1** HAM: Hierarchical Adapters Merging Algorithm

---

**Require:** Task sequence $\mathcal{T} = \{T_1, T_2, \ldots, T_n\}$, base model parameters $\theta_0$, LoRA rank $r$, similarity threshold $\tau$, maximum number of groups $g_{max}$
**Ensure:** Final merged model $\theta_{final}$
1: Initialize adapter set $\mathcal{A} = \emptyset$, scaling factors $\boldsymbol{\alpha} = \emptyset$, group set $\mathcal{G} = \emptyset$
2: **for** each task $T_i$ in $\mathcal{T}$ **do**
3:     Initialize LoRA adapter $\Delta W_i$ with rank $r$
4:     Initialize scaling factor $\alpha_i = 1.0$
5:     **for** each training batch $(x, y)$ from $T_i$ **do**
6:         Compute output: $h = \theta_0(x) + \sum_{j=1}^{i} \alpha_j \cdot \Delta W_j(x)$
7:         Update $\Delta W_i$, $\alpha_i$, and classifier parameters using loss computed from $h$ and $y$
8:     **end for**
9:     Add $\Delta W_i$ to $\mathcal{A}$, $\alpha_i$ to $\boldsymbol{\alpha}$
10:     Assign $T_i$ to an existing group in $\mathcal{G}$ or create a new group based on similarity with previous tasks
11:     **if** number of groups $|\mathcal{G}| > g_{max}$ **then**
12:         Merge the most similar groups in $\mathcal{G}$
13:     **end if**
14:     **for** each group $G_j$ in $\mathcal{G}$ **do**
15:         Compute group adapter: $\Delta W_{G_j} = \text{Concat}\left(\{\alpha_t \cdot \Delta W_t \mid t \in G_j\}\right)$
16:     **end for**
17: **end for**
18: **if** number of groups $|\mathcal{G}| = 1$ **then**
19:     Set $\theta_{final} = \theta_0 + \Delta W_{G_1}$
20: **else**
21:     Merge group adapters using equal weights: $\theta_{final} = \theta_0 + \text{TIES\_Merge}\left(\{\Delta W_{G_j}\}, w = \text{equal}\right)$
22: **end if**
23: **return** $\theta_{final}$

---

the performance gap increasing as the task sequence grows longer, demonstrating superior robustness and scalability. In this scenario, all adapters are merged with the above mentioned strategies after all tasks have been trained.

| Method | 10 Tasks | 20 Tasks | 50 Tasks |
|---|---|---|---|
| TIES | 78.35 ± 1.19 | 78.82 ± 0.14 | 62.53 ± 0.78 |
| LINEAR | 84.75 ± 0.39 | 79.92 ± 1.06 | 64.29 ± 2.98 |
| DARE TIES | 77.76 ± 0.23 | 74.14 ± 0.76 | 62.70 ± 2.04 |
| **HAM** | **86.44 ± 0.30** | **82.07 ± 0.40** | **71.71 ± 2.24** |

Table 10: Comparison of HAM with other model merging baselines on CIFAR-100. HAM yields consistently better performance, particularly on longer task sequences.

## A.5 DECLARATION ON GENERATIVE AI

During the preparation of this work, the authors used OpenAI ChatGPT-4o for grammar and spelling check, paraphrase and reword. After using this tool, the authors reviewed and edited the content as needed and take full responsibility for the publication's content.

