# OpenReview forum: "HAM: Hierachical Adapters Merging for Scalable Continual Learning"
_ICLR.cc/2026/Conference — Submitted to ICLR 2026_

### Official Review · Reviewer_y2TX · 2025-10-30

**Soundness:** 4
**Presentation:** 4
**Contribution:** 3
**Rating:** 6
**Confidence:** 5

**Summary:**

The paper introduces the Hierarchical Adapters Merging (HAM) framework, whose main goal is to mitigate forgetting in ViT models. The proposed method is sound, leveraging the idea of first training task-specific experts (adapters) for each task. Then, it groups the adapters via weight similarity to reduce computational cost. Finally, it merges all grouped adapters into a single module to alleviate the need for task identifiers in class-incremental learning settings. It shows promising results on several datasets compared to previous works.

**Strengths:**

1. Tackle the crucial problem of how to mitigate the forgetting when fine-tuning the transformer-based models.
2. The proposed method is sound and logical. Which uses the concept of MoE and weight similarity to merge the weight, to reduce the computational cost while keeping the critical knowledge for each task.
3. The method shows significant performance gain compared to previous works across 3 datasets.

**Weaknesses:**

1. The testing dataset is limited. In the era of foundation models, testing only on CIFAR/CUB/Tiny-ImageNet seems underwhelming. Please at least add a larger-scale dataset such as ImageNet. Also, it would be much better if the authors could test it on large foundation models such as CLIP and other more complicated tasks, which would show that the proposed method can actually work in real-world use cases.

2. This paper utilizes parameter isolation concept with subsequent weight merging. Since both MoE-like methods [1, 2, 3] and weight merging [4, 5, 6] are established concepts for continual learning, please include citations to these works and provide a concise discussion.

[1] Aljundi, Rahaf, Punarjay Chakravarty, and Tinne Tuytelaars. "Expert gate: Lifelong learning with a network of experts." Proceedings of the IEEE conference on computer vision and pattern recognition. 2017.

[2] Chen, Hung-Jen, et al. "Mitigating forgetting in online continual learning via instance-aware parameterization." Advances in Neural Information Processing Systems 33 (2020): 17466-17477.

[3] Yu, Jiazuo, et al. "Boosting continual learning of vision-language models via mixture-of-experts adapters." Proceedings of the IEEE/CVF Conference on Computer Vision and Pattern Recognition. 2024.

[4] Mallya, Arun, and Svetlana Lazebnik. "Packnet: Adding multiple tasks to a single network by iterative pruning." Proceedings of the IEEE conference on Computer Vision and Pattern Recognition. 2018.

[5] Serra, Joan, et al. "Overcoming catastrophic forgetting with hard attention to the task." International conference on machine learning. PMLR, 2018.

[6] Hung, Ching-Yi, et al. "Compacting, picking and growing for unforgetting continual learning." Advances in neural information processing systems 32 (2019).

**Questions:**

1. The way of grouping adapter weight by just computing cosine similarity is surprisingly simple. Have the authors tried the representation-level similarity (e.g. CKA[7]) instead. It will be better if the authors can provide more qualitative result or analysis on does the grouping works as intended.
2. The concept of merging all trained weights into a single module to alleviate the need for task identifiers is sound. But what if I want to continue training after the merging happens? Will the performance degrade severely since the previously learned knowledge all merges into a single module, which will be brittle? The author should conduct such experiments. This is crucial because if we can’t keep training the model after it is merged, it will defeat the concept of lifelong learning.

[7] Kornblith, Simon, et al. "Similarity of neural network representations revisited." International conference on machine learning. PMlR, 2019.

---

> ### Author Response · Authors · 2025-11-20
> **Rebuttal by Authors**
>
> **Q1 - Beyond cosine similarity**
>
> We thank the reviewer for this insightful observation.
>
> Indeed, the grouping strategy is a crucial component of the HAM methodology, and the choice of the similarity metric is fundamental to its success. We conducted ablation studies to analyze the impact of the grouping rationale, comparing a similarity-based approach against one based on orthogonality. As reported in Table 3 of the manuscript, we found that similarity-based grouping yields a substantial advantage in terms of accuracy, particularly as the number of tasks increases. These results demonstrate that, when integrated with HAM, grouping by similarity is superior, as it effectively helps in consolidating knowledge from multiple tasks and facilitates positive transfer learning within each group.
>
> For this work, we initially selected cosine similarity among the LoRA adapters due to its simplicity, as our primary research focus was on mitigating merging interference across long task sequences. However, we agree that the choice of the metric deserves a deeper investigation. We are currently starting a dedicated set of experiments to explore the behavior of different similarity metrics as an extension of HAM, aiming to fully understand their impact on the method’s overall performance.
>
> **Q2 - Training after merging**
>
> We thank the reviewer for the valuable comment. We agree that the ability to sustain adaptation over an indefinite sequence of tasks is a fundamental area of assessment for true lifelong learning.
>
> Our experimental design, which focuses heavily on extended task sequences (50 and 100 tasks) across fine-grained benchmarks like CUB-200, was specifically chosen to test methods in this more realistic and challenging scenario. We believe that evaluating Continual Learning methods under such rigorous conditions is fundamental for achieving an evaluation closer to real-world applications.
>
> We would like to clarify the role of the final merging phase in the HAM pipeline, as detailed in Section 3.3. This final merging operation is exclusively performed for inference purposes. Its primary objective is to consolidate knowledge into a single unit, thereby significantly **reducing GPU memory footprint** and improving the efficiency of the overall method.\
> However, while GPU memory occupation is paramount for training and inference speed, constraints on disk storage are generally more relaxed, particularly when dealing with small LoRA adapters than occupy few megabytes.
>
> Consequently, our approach is designed to be fully sustainable when a new task arrives. The process for continuous adaptation would be like:
> + **Storage**: The aggregated group adapters are saved on disk. HAM incorporates heavy pruning of the group adapters, which further minimizes the storage requirements
> + **Adaptation**: Upon the arrival of a new task, the current set of group adapters is loaded from disk, taking only few seconds
> + **Execution**: The standard HAM pipeline is then seamlessly executed, incorporating the new task and generating a new set of refined group adapters
>
> Therefore, our method sustains the capacity for continual learning and adaptation without requiring all previous individual adapters. While the issue of optimizing the CL strategy for an infinitely long sequence remains an exciting avenue for future research, HAM represents a robust and scalable solution for the challenges posed by extended task sequences.

---

> ### Comment · Reviewer_y2TX · 2025-11-25
>
> Thanks for clarifying Q2. I agree it’s acceptable to store the weights on disk. However, I keep my score unchanged since the contribution and novelty do not justify raising it higher.
>
> To justify a higher score, the paper would need a clearer novelty statement beyond combining known components, plus stronger evidence on real-world datasets. (w1 and w2)

---

> > ### Author Response · Authors · 2025-11-26
> > **Follow-up rebuttal : Clarifying the performance of HAM on real-world datasets**
> >
> > We thank the reviewer for encouraging us to demonstrate the robustness of HAM on larger, more complex datasets. To address the concern regarding scale (W1), we have conducted new experiments using the ImageNet-R dataset. The new results strongly support our central claim: HAM provides the most balanced and scalable solution for Continual Learning (CL) over long task sequences and potential real-world scenarios.
> >
> >
> > **Accuracy (%) on ImageNet-R 10 & 100 tasks**
> >
> > | Number of tasks | HAM | InfLoRA | TreeLoRA | SD-LoRA |
> > |-----------------|-----|---------|----------|---------|
> > | 10              | 73.45 | 75.52 | 71.94 | 77.37 |
> > | 100             | 49.44 | 29.94 | -*     | 47.73** |
> >
> > - \* The TreeLoRA paper does not provide results on ImageNet-R for more than 10 tasks.
> > - ** The SD-LoRA algorithm fails after 64 tasks due to GPU memory limitations; before the failure, it achieved 47.73%. This means that HAM on 100 tasks outperforms SD-LoRA on much shorter sequences.
> >
> >
> >
> > **Training time (s) on ImageNet-R 10 tasks**
> > | Method | Total time | Average per-task time | Speedup |
> > |--------|------------|----------------------|---------|
> > | SD-LoRA | 5297.02 | 529.70 | 1x |
> > | InfLoRA | 3587.77 | 358.78 | 1.4x |
> > | HAM | 3389.47 | 338.95 | 1.6x |
> >
> >
> > 1. **HAM prioritizes scalability:** Our paper's main contribution is to tackle the scalability of continual learning, especially over long task sequences. This is especially important when thinking of deploying models in real-world scenarios. HAM achieves this by maintaining a constant memory footprint. Unlike SD-LoRA, which fails on many tasks due to memory explosions, HAM is designed to grow gracefully, consolidating knowledge efficiently so it can handle 100 tasks or more without exceeding memory limits.
> >
> > 2. **HAM performance is competitive:** While HAM's 10-task accuracy is competitive, its performance advantage becomes dominant when scalability is tested over long task streams.
> >    * **Long-Sequence Performance:** On the 100-task ImageNet-R sequence, HAM achieved 49.44% accuracy, showing a large performance gap over InfLoRA (29.94%) and proving its ability to mitigate catastrophic forgetting across complex, extended sequences.
> >    * This evidence confirms that the slight trade-off in peak accuracy on very short streams yields a monumental gain in long-term performance preservation and system viability, making HAM the most robust solution for practical CL.
> >
> > 3. **Significant efficiency gains in training:** Beyond just performance, HAM offers substantial speed advantages. Our unique hierarchical merging means that during training, the model operates with reduced, unified adapters, leading to constant and fast training time, regardless of how many tasks it has learned. This translates to roughly twice the speed-up compared to methods like SD-LoRA.
> >
> > 4. **A balanced solution for real-world continual learning:** When we consider the full picture, not just peak accuracy on short task sequences, but also the critical factors of memory usage and speed, HAM presents a superior, more practical solution for real-world continual learning. We've shown a deliberate and effective trade-off: strong, competitive performance combined with a design that guarantees efficiency and scalability for the long haul.
> >
> > When considering the real-world metrics of scalability (100 task completions), efficiency (1.6x speedup), and competitive accuracy, HAM offers a better, more practical solution for Continual Learning. Results on ImageNet-R directly validate that HAM achieves better performance on longer sequences and is less affected by OOD tasks.

---

### Official Review · Reviewer_2Ebs · 2025-11-01

**Soundness:** 2
**Presentation:** 2
**Contribution:** 2
**Rating:** 4
**Confidence:** 3

**Summary:**

This paper proposes HAM (Hierarchical Adapters Merging), a novel continual learning framework that dynamically groups and merges LoRA adapters during training to mitigate catastrophic forgetting and enhance knowledge transfer across tasks. HAM trains task-specific LoRA adapters with importance scalars, clusters similar adapters, prunes less significant weights, and merges them hierarchically. The method is evaluated on three vision benchmarks (CIFAR-100, CUB-200, Tiny-ImageNet) and demonstrates superior performance, especially on long task sequences, achieving up to 4% higher accuracy and nearly double the training/inference efficiency compared to strong baselines.

**Strengths:**

+ Provides a scalable solution for real-world continual learning, especially in long-task scenarios.
+ Propose an interesting dynamically grouping method.

**Weaknesses:**

+ The method is only evaluated in vision tasks; validation in NLP or multimodal settings is missing.
+ The method is not evaluated on commonly used dataset like imagenet-r in pre-trained model based CL. The used CIFAR, CUB, etc. are included in the pre-trained data.
+ During the inference time, existing lora-based cl method can be merged into pre-trained weights to achieve the same inference time as the pre-trained model. However, the proposed method may be not able to achieve this.

**Questions:**

See the weakness

---

> ### Author Response · Authors · 2025-11-20
> **Rebuttal by Authors**
>
> **W1 - Only vision tasks**
>
> We thank the reviewer for the valuable suggestion to broaden the scope of the demonstration. Our strategic decision to prioritize Computer Vision tasks is rooted in the greater intrinsic complexity they show when compared to standard NLP benchmarks. In fact, while NLP tasks benefit from a shared, consistent underlying structure (language, grammar and vocabulary), CV benchmarks show substantial semantic heterogeneity. Consequently, pre-trained models in the CV domain cannot leverage a common structural basis to the same extent as those in NLP.
>
> The contrasting performance of the simplest baselines empirically supports this distinction. Indeed, as indicated in Table 3 of the TreeLoRA paper (suggested by Reviewer mqHm), the FIX baseline (relying purely on in-context learning without any continual fine-tuning) yields results competitive with more sophisticated Continual Learning methods, when applied to NLP tasks. This implies that zero-shot generalization capabilities of LLMs are frequently adequate to achieve strong performance on NLP benchmarks.
> Conversely, the performance of simple baselines in the CV domain is significantly lower than that of competing methods (as demonstrated in Table 1 in HAM and Table 1 in TreeLoRA). This discrepancy underlines that CV tasks are subject to greater inter-task distribution shift and fundamentally demand more powerful and specialized CL methods for effective model adaptation.
> However, we emphasize that this distinction serves to highlight the differing demands of these domains, not to assert the superiority of one benchmark type over the other.
>
> **W2 - No evaluation on ImageNet-R**
>
> We acknowledge the reviewer’s observation regarding the potential for data contamination when evaluating pre-trained models on highly prevalent continual learning datasets, such as CIFAR100 and Tiny-ImageNet. Nevertheless, we want to underscore that our comparative analysis maintains strict methodological consistency by using the same backbone architecture and hyperparameter set for all algorithms assessed. Consequently, any influence or potential bias arising from data overlap is applied uniformly across all the models, thus preserving the integrity and fairness of the performance comparisons.
>
> To rigorously address potential contamination and evaluate performance under out-of-distribution conditions, we incorporated a benchmark that poses a known challenge for many pre-trained networks [1]. Specifically, the CUB-200 dataset, being fine-grained and highly specialized, provides a strong proxy for assessing robustness in novel scenarios. Across this more demanding setting, our proposed HAM approach consistently surpasses all competing techniques, achieving an accuracy of 55.17% compared to 47.56% of SD-LoRA. These results validate the efficacy of our proposed method on out-of-distribution benchmarks and long task sequences.
>
> [1] He, Ju & Chen, Jieneng & Liu, Shuai & Kortylewski, Adam & Yang, Cheng & Bai, Yutong & Wang, Changhu & Yuille, Alan. (2021). TransFG: A Transformer Architecture for Fine-grained Recognition. 10.48550/arXiv.2103.07976.
>
> **W3 - Inference time**
>
> We thank the reviewer for the observation, but we may require further elaboration to address it properly.
>
> The standard procedure for LoRA-based approaches involves integrating the adapter weights into the backbone’s weights, typically performing a weight-wise aggregation. This yields a model with the identical architecture and parameter count as the original backbone, ensuring equivalent inference speed.
>
> HAM pipeline is designed to produce a single, final merged adapter, as detailed in Section 3.3. This final adapter encapsulates all acquired knowledge from the sequence of tasks and can be fully integrated into the backbone’s weights, in the same way as a single LoRA adapter.
> The final merged adapter in HAM is designed to have a higher rank than a single task-specific adapter, which depends on the total number of tasks and the number of groups. In this way, we create a consolidated adapter with sufficient representational capabilities to handle an extended task sequence while maintaining low complexity and a limited parameter count.
>
> Our method strongly differs from similar competitors, such as SD-LoRA, which often require storing and utilising all individual adapters. The key difference is that HAM only requires the storage of k-group adapters, rather than one LoRA per task. This significantly reduces the memory footprint and contributes to the superior efficiency demonstrated in our computational analysis (Figure 3), where HAM is shown to be substantially faster in both training and inference than its best-performing counterparts.

---

> ### Author Response · Authors · 2025-11-26
>
> Dear Reviewer,
>
> We sincerely thank you for the significant time, effort, and thoughtful expertise invested in evaluating our submission.
> We hope the clarifications provided in our rebuttal have fully addressed your concerns. We remain fully available to continue the discussion and promptly answer any remaining questions or points of ambiguity.
> Should you find our response satisfactory, we would be grateful if you would consider revisiting and potentially raising your score.
>
> Thank you again for your crucial feedback.

---

### Official Review · Reviewer_BE3v · 2025-11-03

**Soundness:** 2
**Presentation:** 2
**Contribution:** 2
**Rating:** 2
**Confidence:** 4

**Summary:**

This paper proposes novel methods based on PEFT for continual learning. While there exist a lot of branches of methods for continual learnimg, most of them suffers from lack of plasticity if the network remains same size. Due to this, network expansion should be the most promising way of solving continual learning. However, they lack scalability since network should grow linearly with the number of tasks. In that sense, PEFT naturally gained attention to make network expansion scalable. This paper try to propose new PEFT based extension method which is more efficient than pre-existing algorithms.

**Strengths:**

- Paper is well written and easy to follow. The authors well explained the motivation of each component and the existing related works.
- The authors analyse and disassemble problem well and propose multiple components well aligned with these sub-problems.

**Weaknesses:**

- Although the group number is fixed, if they simply concat the pruned matrices, it is equivalent to concatenating pruned matrices over all tasks. So if this is right, pruning the matrix would be the only component in this method which lacks novelty. Also the authors should compare with this simple approach.

- Backbone is pretrained on ImageNet while the benchmarks seems to be the subset of ImageNet. This can not be fair comparison then.

- Lack of experiments on large scale dataset. Tiny-Imagenet seems to be too small. A lot of existing literatures conducted experiments on at least ImageNet or ImageNet-R scale.

- Lack of comparision with enough baselines. Most of them looks outdated and also the performances are even marginally improved to those.

- Lack of demonstrations on experiments section. There are no demonstrations about how the authors split the task, select the baselines, and which refers to which paper.

**Questions:**

See weaknesses

---

> ### Author Response · Authors · 2025-11-20
> **Rebuttal by Authors**
>
> **W1 - Is HAM only concatenation of pruned matrices?**
>
> HAM indeed includes a concatenation phase involving the LoRA matrices, but **this is only one component of our comprehensive merging pipeline**:
> + First, during the training of task-specific adapters, we concurrently update a set of $\alpha$ values which quantify the relevance of each adapter group (Section 3.2.1). These factors are crucial in the subsequent merging phase (Section 3.3), where they serve as coefficients for the linear merging.
> + Then, by employing a final merging phase rather than simply concatenating all the LoRA matrices, the HAM method's efficiency is greatly improved. In our approach, the number of groups introduces a fundamental trade-off between parameter count (and thus training and inference time) and overall performance. Our experiments confirm that fewer groups yield better results. Nonetheless, even using two groups still achieves a **50% reduction in parameters** compared to naive concatenation of all task-specific matrices.
>
> Another fundamental element of HAM is the dynamic assignment of each new LoRA adapter to a specific group (Section 3.2.2). **Our ablation studies in Section 4.3 clearly demonstrate the significant impact of the grouping strategy on overall performance**. Specifically, comparing similarity-based vs orthogonality-based grouping reveals noticeable differences in accuracy, particularly as the number of tasks grows. This confirms that effective adapter grouping is a crucial component of the overall strategy.
>
> Our approach is not fundamentally restricted to concatenation as the merging algorithm within groups; it is compatible with any combination approach. While we experimented with multiple intra-group merging techniques, e.g., TIES, and found that concatenation provided marginally superior results, the modular nature of this step allows for further investigation into alternative merging methods that could better exploit the similarity of adapters within a group and further improve the efficiency of our approach.
>
> **W2 - No fair comparison**
>
> Two very common and popular benchmarks used in continual learning are CIFAR100 and Tiny-ImageNet. We agree with the reviewer that a potential dataset leakage may exist when using it alongside pretrained backbones. However, we emphasise that the same backbone and hyperparameters were used consistently across all evaluated methods. As such, any potential advantage applies equally, ensuring a fair and controlled comparison.
> To address dataset leakage, we use a benchmark that has been shown to be difficult for most pre-trained backbones [1]. In particular, CUB200 is a fine-grained dataset that helps simulate out-of-distribution scenarios. Even in this setting, HAM outperforms all competing methods, achieving 55.17% accuracy compared to 47.56% for SD-LoRA, demonstrating the robustness of our approach.
>
> [1] He, Ju et al. (2021). TransFG: A Transformer Architecture for Fine-grained Recognition. 10.48550/arXiv.2103.07976.
>
> **W4 - Outdated baselines**
>
> We appreciate the reviewer’s comment regarding the inclusion of baselines. We agree that a comprehensive experimental setup should ideally incorporate the most recent methods. However, in a rapidly evolving field like Deep Learning, the pace of new publications, often coupled with the lack of publicly available codebases, makes it unfeasible to implement and rigorously test every recent approach within the submission timeline.
> The scale of our evaluation further amplifies this challenge. Our experimental design involves three distinct Computer Vision benchmarks, each evaluated across four different sequence lengths (10, 20, 50 and 100 tasks). Given this comprehensive scenario, time and computational cost are constraints.
> Despite these constraints, we selected a heterogeneous and representative set of baselines, encompassing the following categories to provide a robust comparison:
>
> + **Classical CL methods**: Fine-Tuning, EWC and LWF
> + **Merging methods**: Linear Merging, TIES and DARE_TIES
> + **Prompt-based CL methods**: L2P, Dual-Prompt and CODA-Prompt
> + **LoRA-based CL methods**: SD-LoRA, InfLoRA and SEMA
>
> Critically, our selection includes recent state-of-the-art methods, such as SD-LoRA (ICLR 2025), InfLoRA (CVPR 2024) and SEMA (CVPR 2025).\
> The performance results, detailed in Table 1 and Table 2, demonstrate that HAM achieves a noticeable, substantial improvement over the baselines:
> + **Accuracy**: we show an average improvement of approximately 4% in accuracy
> + **Forgetting**: we achieve an average of 9% less forgetting compared to the best competing method, SD-LoRA
>
> Moreover, our analysis of computational efficiency across the top baselines (Figure 3) highlights that HAM is significantly more efficient during both training and inference, being twice as fast as SD-LoRA, the most accurate competitor.
> This demonstrates that HAM not only provides **superior performance** but also offers a compelling **efficiency advantage**.

---

> ### Author Response · Authors · 2025-11-20
> **Rebuttal by Authors**
>
> **W3 - ImageNet-R experiments**
>
> We acknowledge the necessity for further validation on diverse benchmarks. We are currently conducting additional experimental evaluations utilizing the ImageNet-R dataset across varying sequence lengths and will submit these results as a supplemental comment immediately upon completion.
>
> **W5 - No demonstration about the experimental setup**
>
> We thank the reviewer for the comment and apologize for any lack of clarity in the description of our experimental setup.
>
> In Section 4.1, we described the benchmarks used, the evaluation metrics, and the key hyperparameters for both HAM and the competing baselines.
> Regarding the specific concerns:
> + **Task Splitting**: We employ the standard class-incremental learning setup, the most widely adopted and challenging Continual Learning scenario. For each seed, we randomly partition the classes into disjoint groups, each of which is transformed into a new task. As outlined in Section 3.1, our experiments involve a sequence of experiences, each introducing a set of new, previously unseen classes. This configuration is particularly demanding because the model must integrate knowledge about novel labels without having access to the task identifier at inference time. For implementation, we leveraged the Avalanche library, a well-known CL framework that provides robust functionalities for creating such class-incremental scenarios from diverse datasets.
> + **Baseline Selection**: as discussed in our response to the previous point, we designed a heterogeneous and strategically chosen set of baselines for comparison:
>     - Classical CL methods: we included them as fundamental baselines in CL and demonstrated their reduced efficacy when applied to Transformer-based models and long task sequences.
>     - Merging methods: Since HAM’s core objective is to mitigate interference during merging, we included simple merging algorithms to demonstrate that, when unaddressed, merging interference leads to catastrophic forgetting.
>     - State-of-the-art PEFT methods: we compare against well-known, recent CL approaches employing PEFT techniques, both prompt-based and LoRA-based, to prove HAM’s competitiveness.
>
> All the employed baselines are formally introduced, and their respective papers are cited within the Related Works (Section 2) of the manuscript.

---

> ### Author Response · Authors · 2025-11-26
>
> Dear Reviewer,
>
> We sincerely thank you for the significant time, effort, and thoughtful expertise invested in evaluating our submission.
> We hope the clarifications provided in our rebuttal have fully addressed your concerns. We remain fully available to continue the discussion and promptly answer any remaining questions or points of ambiguity.
> Should you find our response satisfactory, we would be grateful if you would consider revisiting and potentially raising your score.
>
> Thank you again for your crucial feedback.

---

### Official Review · Reviewer_mqHm · 2025-11-06

**Soundness:** 2
**Presentation:** 3
**Contribution:** 2
**Rating:** 4
**Confidence:** 4

**Summary:**

The paper proposes Hierarchical Adapters Merging (HAM), a continual learning framework that enhances Low-Rank Adaptation (LoRA) by dynamically grouping and merging task adapters to prevent forgetting and improve knowledge transfer. By assigning importance weights and hierarchically combining related adapters, HAM achieves up to 4% higher accuracy, 9% less forgetting, and twice the efficiency of prior methods, offering a scalable solution for lifelong learning across many tasks.

**Strengths:**

1. The paper is well-structured and clearly written.

2. The study addresses an important problem in continual learning using low-rank adaptation.

**Weaknesses:**

1. The primary concern lies in the novelty of the proposed approach. The idea of Hierarchical Adapters Merging for LoRA appears similar to prior work, such as *TreeLoRA: Efficient Continual Learning via Layer-Wise LoRAs Guided by a Hierarchical Gradient-Similarity Tree*. The authors should clarify the distinctions and emphasize the novel contributions of their method.

2. The paper overlooks a significant body of research on continual learning (or continual fine-tuning) in large language models, including *O-LoRA*, *LoRA-MoE*, and *TreeLoRA*. A discussion comparing this work with these studies would strengthen the paper’s context and positioning.

3. The experimental evaluation is primarily based on synthetic image datasets. It is recommended to include additional experiments on large language models (LLMs) to demonstrate the broader applicability and effectiveness of the proposed method.

References:

[O-LoRA] *Orthogonal Subspace Learning for Language Model Continual Learning*

*LoRA-MoE: Alleviating World Knowledge Forgetting in Large Language Models via MoE-Style Plugin*

*TreeLoRA: Efficient Continual Learning via Layer-Wise LoRAs Guided by a Hierarchical Gradient-Similarity Tree*

**Questions:**

See Weaknesses above

---

> ### Author Response · Authors · 2025-11-20
> **Rebuttal by Authors**
>
> **W1 - Similarity with TreeLoRa**
>
> We appreciate the reviewer for bringing the TreeLoRA paper to our attention. We acknowledge that the temporal proximity of its publication date to our submission deadline meant we were unable to include it in the initial manuscript.
>
> We found several differences between TreeLoRA and HAM. The novelty of our work lies in the following distinctions:
> + **Primary objective**: TreeLoRA primarily aims to enhance the selection of LoRA adapters by utilising a tree-like structure and bandit-based task organisation. In contrast, **HAM addresses the adapter-merging problem** and introduces an **effective technique to reduce merging interference** when combining multiple adapters into a single, cohesive model. Our method integrates various merging techniques with LoRA sparsification to achieve a better balance between knowledge retention and interference.
>
> + **Scope and challenge**: One of the key contributions of our paper is to highlight that many methods proposed in the past years struggle as the number of tasks increases. In contrast, **HAM demonstrates superior performance in long-sequence Continual Learning settings**, handling scenarios with 20, 50, and even up to 100 tasks. We illustrate that this setting poses significant challenges for existing approaches, both in terms of performance and computational resources. HAM's ability to manage interference provides a distinct advantage. Importantly, the authors do not assess TreeLoRA in this more challenging long-sequence context.
>
> + **Empirical performance**: We conducted rigorous testing on fine-grained benchmarks such as CUB-200, which proved notably challenging even for large pre-trained models [1]. In comparing our method to TreeLoRA in the Split-CUB200 scenario with 10 tasks, we found that **HAM achieved an accuracy of 75%, surpassing TreeLoRA's 73%**. This further validates the effectiveness of our merging strategy.
>
> [1] He, Ju et al. (2021). TransFG: A Transformer Architecture for Fine-grained Recognition. 10.48550/arXiv.2103.07976.
>
>
> **W2 - Comparison with O-LoRA and LoRA-MOE**
>
> We appreciate the new references suggested by the reviewer. Although there is a difference between visual and language benchmarks, it is important to highlight the contribution of these works. We will update the paper by adding a short paragraph to highlight the differences between HAM and the suggested references.
>
> The main difference between HAM and previous work is the focus on adding learned information as LoRAs as tasks are learned, while also **reducing interference efficiently**. As we increase the number of tasks, methods that accumulate all adapters (LoRA-MOE, O-LoRA) tend to degrade in performance. On the one hand, an increasing number of adapters slows down training; on the other, interference between adapters increases. HAM addresses this challenge by merging similar adapters during training, keeping training time stable and preventing it from scaling with the number of tasks.
>
> **W3 - Experiments with LLMs**
>
> We appreciate the reviewer’s suggestion to demonstrate broader applicability. We wish to clarify that our experimental evaluation employs widely recognised benchmarks, commonly used by the Continual Learning community. They all consist of photographs with different objects, which are used to determine the classes. For instance, the CUB-200 dataset comprises approximately 12,000 real photographs across 200 fine-grained bird species, representing a challenging scenario even for large-scale pre-trained models.
>
> Our primary focus on computer vision (CV) tasks stems from the greater complexity they pose compared to typical NLP benchmarks. This complexity arises from greater shifts in distribution across various tasks. In contrast to NLP tasks, which share a consistent foundation in natural language structure, vocabulary, and grammar, **CV datasets exhibit significantly greater semantic variance**. This means that pre-trained models in the CV domain cannot rely on the same level of common underlying structure as in NLP.\
> This fundamental difference is evident in the performance of simple baselines. For instance, as shown in Table 3 of the TreeLoRA paper, the FIX baseline (which relies solely on in-context learning without ongoing fine-tuning) performs comparably to more complex continual learning methods for LLMs. This suggests that LLMs' zero-shot capabilities are often sufficient to achieve competitive performance on NLP benchmarks. In contrast, **the performance of naive baselines in the CV domain is notably poor**, as illustrated in both our study (Table 1) and in the TreeLoRA paper (Table 1). This confirms that **CV tasks are inherently more challenging** in terms of distribution shift between tasks and require a different set of continual learning strategies to adapt the model effectively.
>
> It is important to note that we are not suggesting that one type of benchmark is better than another; rather, they require different approaches.

---

> > ### Comment · Reviewer_mqHm · 2025-11-25
> >
> > Thanks to the author for the response, which partially addressed my concerns. I have therefore decided to raise my score.

---

### Author Response · Authors · 2025-11-21
**Rebuttal - New experiments on ImageNet-R**

After receiving the reviews, we conducted some experiments on the ImageNet-R dataset.
We provide in the next table the results we obtained on different numbers of tasks.

**Accuracy (%) on ImageNet-R, 10 & 100 tasks**

| **Number of tasks** | **HAM** | **InfLoRA** | **TreeLoRA** | **SD-LoRA** |
|---------------------|---------|-------------|--------------|-------------|
| 10                  | 73.45   | 75.52       | 71.94        | 77.37       |
| 100                 | 49.44   | 29.94       | -*           | 47.73\**    |

\* The TreeLoRA paper does not provide results on ImageNet-R for more than 10 tasks.\
** The SD-LoRA algorithm fails after 64 tasks due to GPU memory limitations; before the failure, it achieved 47.73%. This means that **HAM on 100 tasks outperforms SD-LoRA on much shorter sequences.**

**Training time (s) on ImageNet-R for 10 tasks**

| **Method** | **Total time** | **Average per-task time** | **Speed-up** |
|------------|----------------|---------------------------|--------|
| SD-LoRA    | 5297.02        | 529.70                    | 1x           |
| InfLoRA    | 3587.77        | 358.78                    | 1.4x         |
| HAM        | **3389.47**    | **338.95**                | **1.6x**     |

Based on these empirical results, our key findings are as follows:

+ **HAM prioritizes scalability**: Our paper's main contribution is to tackle the scalability of continual learning, especially over long task sequences. HAM achieves this by maintaining a constant memory footprint. Unlike SD-LoRA, which fails on many tasks due to memory explosions, HAM is designed to grow gracefully, consolidating knowledge efficiently so it can handle 100 tasks or more without exceeding memory limits.

+ **HAM performance is competitive**: While the ImageNet-R results show other baselines with slightly higher scores, HAM is very much in the same league. In particular, HAM outperforms other competitors (InfLoRA and SD-LoRA) in longer sequences (100 tasks). This gap highlights that HAM performs well, demonstrating its ability to learn complex tasks effectively, especially when dealing with an higher number of tasks.

+ **Significant efficiency gains in training**: Beyond just performance, HAM offers substantial speed advantages. Our unique hierarchical merging means that during training, the model operates with reduced, unified adapters, leading to constant and fast training time, regardless of how many tasks it has learned. This translates to a significant speed-up compared to methods like SD-LoRA.

+ **A balanced solution for real-world continual learning**: When we consider the full picture, not just peak accuracy on long task sequences, but also the critical factors of memory usage and speed, HAM presents a superior, more practical solution for real-world continual learning.
Our approach demonstrates a deliberate and effective trade-off, achieving strong, competitive performance alongside a design that guarantees enhanced efficiency and long-term scalability.

---

### Author Response · Authors · 2025-11-28
**General Response to all Reviewers**

We sincerely thank the reviewers for their insightful comments, which recognized the importance of evaluating Continual Learning methods in **extended, more realistic scenarios**.

As highlighted by **Reviewers y2TX, 2Ebs and BE3v**, our Hierarchical Adapters Merging structure is technically sound. We have thoroughly analyzed each subcomponent, providing numerous insights and ablation studies in the manuscript.
Our dynamic grouping technique (acknowledged as interesting by **Reviewer 2Ebs**) represents the main technical contribution which, combined with the strategic management on the importance factors $\alpha$, yields **superior results** across various benchmarks while simultaneously providing **extremely higher computational efficiency** than competing state-of-the-art methods.

We have carefully addressed the points raised in the initial reviews.
Specifically, we provided comprehensive clarifications to **Reviewer BE3v** regarding our methodology, demonstrating the significant impact of our overall grouping strategy on both performance and efficiency.
Additionally, we performed **further experiments on ImageNet-R** , which clearly show that our method surpasses competing baselines,
particularly in longer and more complex CL scenarios.
Furthermore, these new results reconfirm **HAM’s massive speedup**, establishing it as a **highly scalable solution for real-world continual learning applications**.

As the rebuttal period draws to a close, we remain fully available for any further discussion with the reviewers. Based on our answers and the supporting experimental evidence, we would be genuinely grateful if the reviewers would reconsider their comments and potentially revise and raise their scores.

Thank you once again for your indispensable feedback.

---

### Author Response · Authors · 2025-12-03
**Global Response**

We sincerely appreciate the thorough and constructive points raised by all four reviewers, and we deeply thank them for their time and expertise. During the rebuttal period, we have comprehensively responded to all comments, providing detailed methodological clarifications and reporting the results of newly conducted experiments.
Below, we synthesize the major revisions undertaken and the definitive outcomes established through the discussion phase.

# Summary of Responses
We report in the following the answers we provided during the rebuttal stage, to address reviewers’ main concerns:

+ **Additional experiments on ImageNet-R**:  We conducted and reported further experiments on the ImageNet-R dataset, specifically focusing on challenging, longer task sequences. These results unequivocally demonstrate that **our method surpasses competing baselines** in complex Continual Learning scenarios. Crucially, these new experiments definitely reconfirm **HAM’s massive speedup**, establishing it as a highly scalable and practical solution for real-world Continual Learning applications. (Response to Reviewers BE3v, 2Ebs and y2TX)

+ **Clarification on Domain Focus (CV vs NLP)**: In response to Reviewers mqHm and 2Ebs, we clarified the strategic rationale behind our focus on Computer Vision tasks. Our experimental setup, based on real-world images from widely recognized benchmarks, was chosen due to the **greater semantic variance and inter-task distribution shifts** inherent to CV compared to NLP benchmarks. We substantiated this choice by referencing external literature, such as the TreeLoRA paper [1] pointed out by mqHm, which independently highlights the differing baseline efficacy across these domains.

+ **Methodological Originality and Efficacy**: Reviewers mqHm and BE3v raised questions regarding specific elements of the HAM methodology. We provided detailed clarifications about our hierarchical merging procedure and the distinction from existing methods. We demonstrated the **clear efficacy of our dynamic grouping strategy** through ablation studies on both sheer performance gains and computational efficiency, explicitly referencing the relevant sections in the manuscript.

[1] Qian, Yu-Yang, et al. "TreeLoRA: Efficient Continual Learning via Layer-Wise LoRAs Guided by a Hierarchical Gradient-Similarity Tree." Forty-second International Conference on Machine Learning.

# Status of Reviewer Responses
+ **Reviewers who raised the score**: Reviewer mqHm (Initial score: 4; Post-rebuttal: 6). The reviewer expressed satisfaction with our detailed clarifications on the paper’s originality and our justification for focusing on the greater complexity on CV benchmarks, leading to an increase in the score.
+ **Reviewers maintaining a high score**: Reviewer y2TX (Initial score: 6). The reviewer acknowledged our responses regarding disk usage and requested additional experiments on larger benchmarks (ImageNet-R). We have provided these additional results, confirming HAM’s superior performance and efficiency in long sequences. Although the reviewer did not engage further following the submission of this new data, we are confident that our comprehensive response fully addressed the remaining concerns.
+ **Reviewers who have not responded**:
    + *Reviewer BE3v* (Initial score: 2)
We provided detailed responses to all methodological and experimental concerns, including the additional experiments on ImageNet-R. We believe that these answers, clearly highlighting the originality and effectiveness of HAM, fully resolve the issues raised.

    + *Reviewer 2Ebs* (Initial score: 4)
        We addressed the reviewer’s concerns regarding LLM experiment justification and inference time complexity, providing methodological clarifications. We are confident that our responses, together with the new results on ImageNet-R, successfully resolve the doubts about the model’s efficacy and efficiency in complex, longer CL scenarios.

We want to explicitly thank all reviewers for their valuable comments and their significant efforts. We believe our responses and the supporting evidence have successfully demonstrated the novelty, efficacy and scalability of our HAM method. We hope the final judgment will reflect these substantial improvements and clarifications.

---

### Meta-Review · Area_Chair_Zi1C · 2026-01-12

**Summary:**

Reviewers broadly agree the submission addresses an important practical problem in continual learning: scaling PEFT/LoRA approaches to long task streams without memory/time blowing up. The main concerns driving the initial mixed scores were
- novelty (the major concern of mine, too) whether HAM is meaningfully new versus existing hierarchical/grouping/merging LoRA approaches (e.g., TreeLoRA) or even a simpler prune + concatenate baseline
- early reviews found CIFAR/CUB/Tiny-ImageNet underwhelming for foundation-model-era CL and asked for ImageNet(-R)/more realistic settings
- how tasks are split, how baselines were selected/implemented, and whether pretraining overlap could confound comparisons

**Reviewer Concerns:**

Addressed:
- the authors provided detailed clarifications (task splitting, baseline categories, inference-time behavior)
- added ImageNet-R results targeting long sequences, which materially reduces the too small / not realistic enough concern.

Outstanding:
- novelty remains the key residual disagreement
- The rebuttal justifies focusing on CV as harder/more distribution-shifted than typical NLP benchmarks, but the absence of any NLP/VLM/LLM experiments still limits generality

**Reviewer Scores:**

Likely all remain the same (4, 2, 4, 6)

---

### Decision · Program_Chairs · 2026-01-26

Reject